# Multicriteria Optimization Problem on Prefractal Graph

Rasul Kochkarov 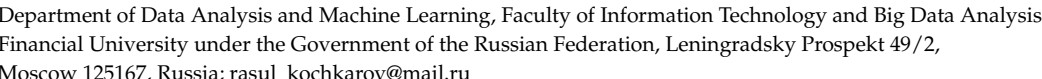

Department of Data Analysis and Machine Learning, Faculty of Information Technology and Big Data Analysis, Financial University under the Government of the Russian Federation, Leningradsky Prospekt 49/2, Moscow 125167, Russia; rasul_kochkarov@mail.ru

**Abstract:** Even among single-criteria discrete problems, there are NP-hard ones. Multicriteria problems on graphs in many cases become intractable. Currently, priority is given to the study of applied multicriteria problems with specific criteria; there is no classification of criteria according to their type and content. There are few studies with fuzzy criteria, both weight and topological. Little attention is paid to the stability of solutions, and this is necessary when modeling real processes due to their dynamism. It is also necessary to study the behavior of solution sets for various general and individual problems. The theory of multicriteria optimization is a rather young branch of science and requires the development of not only particular methods, but also the construction of a methodological basis. This is also true in terms of discrete graph-theoretic optimization. In this paper, we propose to get acquainted with multicriteria problems for a special class of prefractal graphs. Modeling natural objects or processes using graphs often involves weighting edges with many numbers. The author proposes a general formulation of a multicriteria problem on a multi-weighted prefractal graph; defines three sets of alternatives—Pareto, complete and lexicographic; and proposes a classification of individual problems according to the set of feasible solutions. As an example, we consider an individual problem of placing a multiple center with two types of weight criteria and two types of topological ones. An algorithm with estimates of all criteria of the problem is proposed.

**Keywords:** multicriteria problem; prefractal graph; set of alternatives; algorithm with estimates

## 1. Introduction

### 1.1. Single-Criteria Optimization Problems

Classical optimization problems on graphs are represented by one criterion. The essence of the problem is the selection of the required optimal subgraph according to a given criterion. This is the selection of spanning subgraphs (trees, chains, cycles, and other structures) of minimum or maximum weight, the search for optimal paths, maximum matchings, minimum cost flows, placement of centers and medians, allocation of Hamiltonian and Euler subgraphs, and the coloring of vertices and edges [1–3]. In the formulation of the optimization problem, a set of feasible solutions is determined—all possible subgraphs of the required structure, and among them the optimal solution according to a given criterion is selected. There may be several optimal solutions, but they all lead to an equal criterion value. Statements of modern single-criterion problems correspond to classical statements, while they contain particular refinements related to practical application in a particular branch of research. There is also a bias towards new methods and algorithms for solving known problems. In [4], the problem of coloring large graphs is studied and Memetic Teaching-Learning-Based optimization algorithms are proposed in serial and parallel implementations. In [5], the Honeybee optimization algorithm for the graph K-decomposition problem was proposed. To solve the multivariate traveling salesman problem (MTSP) in [6], a learning-based approach is proposed. This is a new formulation of the classical combinatorial traveling salesman problem. Optimization problems also include problems of covering a graph with intersecting or non-intersecting structures. In the case of non-intersecting structures, we can talk about the selection of non-intersecting

subgraphs. In [7], the problem of decomposing a graph into stars of minimum size is studied. It should be noted that the statement of this problem belongs to the class of NP-hard problems, and many modern statements, due to the complication of criteria and additional conditions, are also intractable [8].

### 1.2. Multicriteria Optimization

The field of study of multicriteria problems on graphs is not new [9–13]. The works of those times are mainly represented by multicriteria problems on graphs weighted by many weights. For each set of weights, a separate criterion is set and various methods are proposed for finding the optimal solution for each criterion.

Despite significant publications, there are still no universal methods for solving multicriteria problems on graphs. In [14], a multicriteria graph problem with criteria of the MAXMIN type was studied and an algorithm for finding a solution from a complete set of alternatives was proposed. In [15], two methods for solving the multicriteria problem of finding the shortest path on a rough graph (a graph with nondeterministic edge weights) were proposed: a modified Dijkstra algorithm and a rough programming method. It should be noted that even classical problems on large and complex graphs require new approaches to finding solutions. Thus, a new algorithm is proposed for computing Pareto-optimal shortest paths on a network weighted by fuzzy weights [16]. The study also paid attention to the fuzziness of the criteria themselves, which adds additional complexity to finding a solution. To study the sets of non-dominated decisions, classical interactive methods with the participation of the decision-maker are adapted. As a numerical experiment, the proposed method was applied to solve multicriteria problems of finding the shortest path and spanning tree [17]. Of course, the theory of multicriteria optimization is becoming more and more relevant, in particular, with regard to large graphs and networks with many weights and complex connection structures [18,19], including for solving applied model problems [20]. To solve problems of multicriteria optimization, a graph-theoretic approach can be used, for example, in the problem of splitting a social graph in real-time [21] or transport and logistics problems [22,23].

### 1.3. Related Works

In the formulation of a multicriteria problem, there are several criteria (two or more), and a set of feasible solutions (a set of alternatives) is specified. Determining the set of alternatives, its structure, as well as the selection of the entire set of alternatives, that is, the enumeration of all possible subgraphs, is a separate task, and often intractable [24–26]. Then, from the set of alternatives, the optimal solution according to the criteria is selected. The question then arises as to which solution to choose as optimal in the case of many opposite criteria. To select the optimal solution to a multicriteria problem, various methods have been developed—Pareto selection (Slater, Nash, etc.), lexicographic order of criteria [27], or various criteria convolution methods (scalarization, normalization, weighted sums, etc.) [28].

The application of the theory of multicriteria optimization is accepted in various scientific and applied fields [29]. In this paper, attention is paid to problems on prefractal graphs in the terminology and notation in [30–32]. An introduction to the class of prefractal graphs is given in the author's papers [33–37]. The following are the main definitions and notations of a prefractal graph used in this paper.

A prefractal graph is denoted as $G_L = (V_L, E_L)$, where $V_L$ is the set of vertices, and $E_L$ is the set of edges. In what follows, a simplified notation $G_L$ is used for known (canonical) prefractal graphs $G_L = (V_L, E_L)$. In the process of constructing a prefractal graph, a trajectory is formed $G_1, G_2, \ldots, G_L$. The graph constructed at step $l = 1, 2, \ldots, L$ is called a prefractal graph $G_l$ of rank $l$. The new edges of the graph $G_L$ are the edges of rank $L$, and the remaining edges are the old edges of the rank $l$. As $l \to \infty$, the graph $G_l$ is fractal. A fractal graph, like a fractal, is an infinite object. For a fixed value of rank $l$, a prefractal graph is considered. For example, as shown above for $l = L$ the graph $G_L$ is considered.

Essential characteristics of a prefractal graph $G_L = (V_L, E_L)$ are the number of its vertices in Equation (1) and edges in Equation (2).

$$N = N(n, L) = |V_L| = n^L, \tag{1}$$

where $n = |W|$ is the number of vertices of seed $H$.

$$M = M(n, q, L) = |E_L| = q\left(1 + n + n^2 + \cdots + n^{L-1}\right) = q\left(n^L - 1\right)/(n-1), \tag{2}$$

where $q = |Q|$ is the number of edges of seed $H$.

Thus, in this paper, we propose a toolkit of prefractal graphs for formulating multicriteria problems. A rule for weighing a prefractal graph by many real numbers is proposed, and classes of individual problems with common sets of feasible solutions are presented as an example. Two types of criteria are proposed—topological and weight.

## 2. Multicriteria Problem on Prefractal Graph

### 2.1. Set of Alternatives

A common problem of discrete multicriteria problems is finding sets of alternatives. The questions of estimating the complexity of finding sets of alternatives, the effectiveness of algorithms—exact and approximate—are considered. In this paper, attention is paid to finding at least one optimal solution from a set of alternatives. Let us present the definitions and notation used in the description of multicriteria problems [34] as applied to prefractal graphs.

The set of alternatives (SA) is a set of all possible alternative solutions to the problem, including optimal and near-optimal solutions. The concept of SA is primary and is introduced for the needs of the theory of choice and decision making. In practice, SA is subject to conflicting requirements: it must be the most representative, including all the «best» and close to the solutions, and at the same time must be visible to the decision-maker (DM). The concept of SA appeared as a result of the need to make a decision under the conditions of several simultaneous criteria. In this case, a situation arises of the existence of alternative solutions, each of which is better than the other in at least one criterion. Similarly to single-criteria optima, one speaks of «multicriteria optima», which are found in the literature [38] under the names of Pareto-optimal, efficient, non-dominated, etc. solutions. However, in contrast to single-criteria optimization, finding one specific multicriteria optimum is not easy even for a particular problem.

The process of finding the SA should be completed by the representation of the elements in one form or another. In the theory of choice and decision making, the most common are three ways:

(1)   explicit listing of all competing alternatives;
(2)   representation of SA elements in an implicit form with the help of additional systems of restrictions;
(3)   construction of a deterministic formal mechanism that allows for the generating of alternatives.

The mathematical formulation of a discrete multicriteria problem consists of a description of the conditions that determine a finite or countable set of feasible solutions $X = \{x\}$ and a vector-objective function (VOF) defined on this set:

$$\begin{aligned} F(x) &= (F_1(x), F_2(x), \ldots, F_i(x), \ldots, F_M(x)), \\ F_i(x) &\to extr. \end{aligned} \tag{3}$$

It is customary to speak of an *individual problem* (or *model problem*) if all parameters of the vector-objective function and a system of constraints describing the set of feasible solutions (SFS) are given. If some of these parameters are not fixed but are represented by generally accepted notations, it is customary to talk about a *mass* problem (from the

word *massive*), or briefly, about a problem. Examples of mass problems are the traveling salesman problem, the Euler graph problem, the transportation problem, etc.

The numerical solution of an individual problem is to find the SA $X^* \in X$ from the SFS. In a broad sense, the solution of a problem is understood as the construction of a certain algorithm that guarantees the finding of an SA for any individual problem of a mass problem.

How to evaluate the complexity of finding SA? In this paper, an algebraic approach is used to estimate the computational complexity, measured by the number of required arithmetic operations. This does not take into account the representation of a discrete data structure in a computer, so there is no need to investigate the performance of operations in a particular machine and the cost of the digital length of numbers. However, due to the variety of computing platforms, additional clarifications are introduced if necessary. At the same time, for parallel machines, the concept of time complexity means the greatest total time (measured by arithmetic operations) spent by one of the parallel processes.

The paper uses the concept of asymptotic time complexity—the behavior of computational complexity as a function of the size of the input in the limit as the size of the problem increases. Estimating the complexity in the worst case, we use the existing hierarchy of the form: «polynomial problems»-«NP-complete problems»-«intractable problems». The complexity for almost all individual problems is an upper bound on the complexity of the mass problem.

Returning to the sets of alternatives, three types of SA are considered, each of which is a proper or improper subset of the Pareto set $\widetilde{X}$.

The *Pareto set (of alternatives)* (PSA) $\widetilde{X}$ consists of all Pareto-optimal solutions. For a given individual problem with a vector-objective function (3), an element $x^0 \in X$ is called Pareto-optimal (non-dominated) if there is no such element $x^* \in X$ that satisfies the inequalities $F_i(x^*) \leq F_i(x^0)$, $i = \overline{1, M}$, among which at least one is strict. The Pareto principle says that only the element that belongs to the PSA should be chosen as a solution to the problem. It follows from the Pareto principle that PSA is the most representative type of SA of maximum power. In the single-criteria case, the PSA is the set of all optima of the problem under consideration.

The complete set of alternatives (CSA) is a subset $X^0 \subseteq \widetilde{X}$ of minimal cardinality such that $F(X^0) \subseteq F(\widetilde{X})$. If for an individual problem the PSA and the SCA do not coincide, then the SCA is not uniquely determined and there are at least several different choices for the SCA for this problem.

The *lexicographic set of alternatives* (LSA) $X_\Lambda^0$ is defined as follows. Finding any one lexicographic optimum is adequate to such a formulation of a multicriteria problem in which the criteria are ordered by importance and numbered so that each previous one is incomparably more important than all subsequent ones. Then one speaks of a lexicographic problem, the essence of which is to achieve an arbitrarily small improvement in an important criterion at the expense of arbitrarily large losses in all other less important criteria. Finding the LSA in some cases is an easier problem than finding the CSA. In the case of using common methods—linear convolution algorithms, then in the class of these algorithms the problem of finding the CSA is not solvable, while the problem of finding the LSA is solvable. In the context of algorithmic problems of discrete optimization, the LSA is one of the possible approximations of the desired CSA. Formally, LSA is described as follows. Let $\Lambda = \{\lambda\}$ be the set of all $n!$ permutations of numbers $1, 2, \ldots, n$. An element $x' \in \widetilde{X}$ is called a lexicographic optimum if there is such a permutation $\lambda = i_1, i_2, \ldots, i_n$ in $\Lambda$, that for each $x \in \widetilde{X}$ one of the following two conditions is satisfied:

(a)  $F_i(x') = F_i(x)$, $i = 1, 2, \ldots, n$;
(b)  there exists $k \in \{1, 2, \ldots, n\}$ such that $F_{i_k}(x') < F_{i_k}(x)$ and $F_{i_r}(x') = F_{i_r}(x)$, $r = 1, 2, \ldots, k - 1$.

Let $\widetilde{X}_\Lambda$ be the set of all lexicographic optima defined on $\Lambda$, $\widetilde{X}_\Lambda \subseteq \widetilde{X}$. Then the LSA is a subset $X_\Lambda^0 \subseteq \widetilde{X}_\Lambda$ of minimal cardinality such that $F(X_\Lambda^0) = F(\widetilde{X}_\Lambda)$. Any LMA $X_\Lambda^0$ can

be defined as the intersection of a certain CSA $X^0$ with $\widetilde{X}_\Lambda$: $X^0_\Lambda = X^0 \cap \widetilde{X}_\Lambda$. At the same time, if for an individual problem the PSA and the LSA do not coincide, then the LSA is not uniquely determined and there are at least several different choices for the LSA.

### 2.2. Classification of Multicriteria Problems on Prefractal Graphs

An analysis of the descriptions of discrete graph problems suggests that the composition of the criteria for a vector-objective function usually changes from one individual problem to another. For example, a spanning tree in an optimization problem can be evaluated by the criteria of weight, degree, and diameter, the «capacity» of edges, etc. Thus, there are various variants of individual spanning tree problems. What these problems have in common is the definition of a set of feasible solutions. For this reason, the mathematical formulation of a multicriteria problem of any kind seems to be similar to the description of the set of feasible solutions. After that, the criteria for individual tasks are concretized and algorithms for finding their solutions are given.

Speaking about an individual problem on a prefractal graph (digraph) $G_L = (V_L, E_L)$, we assume that its feasible solution is a subgraph $x = (V_x, E_x)$ with a set of vertices $V_x \subseteq V_L$ and a set of edges (arcs) $E_x \subseteq E_L$. In some cases, an admissible solution to a digraph can be supplemented with clarifying conditions. Here are some definitions and notations: $G_L^N$ is the set of all $N$-vertex prefractal graphs $G_L$; $G_L^{N,M}$ is the set of all $N$-vertex $M$ weighted prefractal graphs $G_L$.

Individual multicriteria problems on multi-weighted (with real numbers) prefractal graphs will be denoted by:

$Z_1$—the problem of multiple centers, $x = (V_x, E_x)$—$p$-center of $G_L$;

$Z_2$—the problem of multiple medians, $x = (V_x, E_x)$—$p$-median of $G_L$;

$Z_3$—the problem of spanning forests, $x = (V_L, E_x)$—spanning forest of $G_L$;

$Z_4$—the problem of perfect matchings, $x = (V_L, E_x)$—perfect matching of $G_L$, $x = (V_L, E_x)$—perfect matching of $G_L$, and $|V_L|$ is an odd number, $x$ is the maximum matching and $|E_x| = (N-1)/2$;

$Z_5$—the problem of perfect matchings on a bipartite graph, $x = (V_L, E_x)$—perfect matching of $G_L = \left( V_L^{(1)}, V_L^{(2)}, E_L \right)$, and $\left| V_L^{(1)} \right| \neq \left| V_L^{(2)} \right|$ is an odd number, $x$ is the maximum matching and $|E_x| = \min\left\{ \left| V_L^{(1)} \right|, \left| V_L^{(2)} \right| \right\}$;

$Z_6$—the problem of the shortest chains (paths) between a pair of vertices, $x = (V_x, E_x)$—shortest simple chain (path) between two given vertices $v_1, v_2 \in V_L$ of a prefractal graph (digraph) $G_L$;

$Z_7$—the traveling salesman problem, $x = (V_L, E_x)$—Hamiltonian cycle (contour) of a prefractal graph (digraph) $G_L$;

$Z_8$—the Euler cycle covering problem, $x = (V_L, E_x) = \{C_m\}$ is a spanning subgraph of $G_L$, each component $C_m$ of which are an Euler graph and the components in the covering do not intersect;

$Z_9$—the problem of covering a graph with chains, $x = (V_L, E_x)$ is a spanning subgraph of $G_L$, each connected component of which is a $h$-chain, $2 \le h \le h_{max}$;

$Z_{10}$—the problem of covering a graph with stars, $x = (V_L, E_x)$ is a spanning subgraph of $G_L$, each connected component of which is $h$-stars, $2 \le h \le h_{max}$;

$Z_{11}$—the vertex coloring problem, $x = (V_x, E_x)$ is the correct coloring (partitioning) of the set of vertices $V_L$ of $G_L$, $V_x = \left\{ V_x^1, V_x^2, \ldots, V_x^i, \ldots, V_x^k \right\}$, $k$ is the number of colors in the coloring $x$.

Classification of problems $Z_t$, $t = 1, 2, \ldots, 11$ is called classification according to the type [39]. Such a classification is generally recognized and defines problems by the set of feasible solutions for each type. The classification can be extended by known individual problems.

For problems on prefractal graphs, another type of classification is applicable, based on the method of constructing a prefractal graph. On a prefractal graph, you can process

individual subgraphs, and then combine the results in a different configuration depending on the criteria of the problem. Subgraph-seeds and blocks of different ranks are considered subgraphs. Blocks of the $r$ rank are connected subgraphs of the prefractal graph obtained by removing all edges of rank $l = 1, 2, \ldots L - r$. Blocks of the first rank are called seeds of the prefractal graph. The classification of problems according to their structural affiliation is carried out as follows.

$Z_{l_1}^{l_2}$—the problem of covering seeds of rank $l_1, l_1 + 1, \ldots, l_2$ of a prefractal graph $G_L$. In particular, $Z_1^L$ is the problem of covering seeds from the first to the $L$ rank inclusive, which corresponds to covering the entire prefractal graph $G_L$.

$B_{l_1}^{l_2}$—the problem of covering blocks of rank $l_1, l_1 + 1, \ldots, l_2$ of a prefractal graph $G_L$. In particular, $B_L^L$ is the problem of block coverage of rank $L$, which corresponds to the coverage of the entire prefractal graph $G_L$.

*2.3. Complete Problems and Their Sets of Alternative Solutions*

Problem $Z_t$ implies the set of all of its individual problems. The set $X_t = \{X\}$ denotes the set of feasible solutions to a problem $Z_t$, obtained by combining the sets of feasible solutions of all its individual problems. Individual problems of one family $Z_t$, $t = 1, 2, \ldots, 11$ have the same definitions of a feasible solution but differ in the dimension of the vector-objective function, the composition of the criteria, the number of sets of weights, etc.

A multicriteria problem $Z_t$ is called a *complete problem* if for each set of feasible solutions $X \in X_t$ there are such parameters of its vector-objective function that the equality $X^0 = \widetilde{X} = X$ is true. For prefractal graphs, the following lemmas of the theory of multiobjective optimization are true.

**Lemma 1.** *For any problem with a vector-objective function of the form $F : X \to R^N$, the cardinalities are equal $\left|X^0\right| = \left|F\left(\widetilde{X}\right)\right|$, where $R^N$ is the Euclidean space of finite dimension $N$.*

**Lemma 2.** *Adding new criteria to the vector-objective function of any individual problem either leaves its PSA and CSA unchanged or replenishes them with new alternatives.*

**Lemma 3.** *For fixed $t$, some individual problems from $Z_t$ can be complete, while other problems of the same family do not have the completeness property.*

The Lemmas 1–3 proposed above and the following Theorems 1–6 are true for multicriteria problems on graphs, including the class of prefractal graphs. All required proofs are given in [34].

**Theorem 1.** *Any spanning forest problem $Z_3$ is complete if its vector-objective function contains at least two weight criteria.*

**Theorem 2.** *Any perfect matching problem $Z_4$ is complete if its VOF contains at least two weight criteria.*

**Theorem 3.** *Any perfect matching problem $Z_5$ on a bipartite prefractal graph is complete if its VOF contains at least two weight criteria.*

**Theorem 4.** *Any traveling salesman problem $Z_7$ is complete if its VOF contains at least two weight criteria.*

The completeness of the problem means that any found solution from the set of feasible solutions will be Pareto and at the same time be included in the complete set of alternatives. Moreover, singling out the entire set of alternatives means singling out the entire Pareto set.

Theorems 1–4 give completeness conditions for problems under which any solution from the set of alternatives can be found, and it will be Pareto.

The remaining problems $Z_t$ require a separate analysis to identify the completeness property.

**Theorem 5.** *The problem of finding the PSA $\widetilde{X}(CSA\ X^0)$ of a typical problem $Z_t$ on a multi-weighted prefractal graph ($M \geq 2$) is unsolvable using linear convolution algorithms.*

Theorem 5 postulates the fact that in the case of 2 or more weights, the problem is unsolvable using linear convolution algorithms. In this sense, it is necessary to consider weights of a different nature, incomparable among themselves.

Nevertheless, linear convolution algorithms can be used to find nontrivial SA, in particular LSA.

**Theorem 6.** *The problem of finding the LSA of a typical integer problem $Z_t$ under the condition of only weight criteria on a multi-weighted prefractal graph ($M \geq 2$) is solvable using linear convolution algorithms.*

*2.4. Formulation of a Multicriteria Problem on a Multi-Weighted Prefractal Graph*

We consider a prefractal graph $G_L = (V_L, E_L)$ generated by a set of seeds $H = \{H\}$. Each edge of $l$ rank $e^{(l)} \in E_L$ is weighted by $M$, $M \geq 1$ real numbers $w_i\left(e^{(l)}\right) \in \left(\theta^{l-1}a, \theta^{l-1}b\right)$, $i = \overline{1,M}$, where $l = \overline{1,L}$, $a, b > 0$, $a < b$ and $0 < \theta < a/b$—similarity coefficient $\theta \in (0,1)$.

On the set of feasible solutions $X = X(G_L) = \{x\}$: $x = (V_x, E_x)$, $V_x \subseteq V_L$, $E_x \subseteq E_L$, the vector-objective function is given:

$$F(x) = (F_1(x), F_2(x),\ \ldots, F_i(x), \ldots, F_M(x), \ldots, F_{M+T}(x)), \tag{4}$$

in which the criteria:

$$F_i(x) = op[\omega_i(e)] \to extr,\ i = 1,2\ldots M, \tag{5}$$

$$F_i(x) = op[\psi_i(v,e)] \to extr,\ i = M+1, M+2, \ldots, M+T, \tag{6}$$

where $op$ is an operation on a function of the form *max, min, sum*.

Criteria $F_i(x)$, $i = \overline{1,M}$ from Equation (5) are called *numerical* (*weight*) *criteria* and are functions of weight characteristics, and criteria $F_i(x)$, $i = \overline{M+1, M+T}$ from Equation (6)—*topological*, constructed based on topological (structural) characteristics of the prefractal graph.

Weight criteria can take the form:

$$
\begin{aligned}
F_i(x) &= \max_{e \in E_x}\omega_i(e) \xrightarrow{x} min, \\
F_i(x) &= \min_{e \in E_x}\omega_i(e) \xrightarrow{x} max, \\
F_i(x) &= \textstyle\sum_{e \in E_x} \omega_i(e) \xrightarrow{x} min, \\
F_i(x) &= \textstyle\sum_{e \in E_x} \omega_i(e) \xrightarrow{x} max.
\end{aligned}
\tag{7}
$$

Topological criteria can take the form:

$$
\begin{aligned}
F_i(x) &= \max_{e \in E_x, v \in V_x} \psi_i(v,e) \xrightarrow{x} min, \\
F_i(x) &= \min_{e \in E_x, v \in V_x} \psi_i(v,e) \xrightarrow{x} max, \\
F_i(x) &= \textstyle\sum_{e \in E_x, v \in V_x} \psi_i(v,e) \xrightarrow{x} min, \\
F_i(x) &= \textstyle\sum_{e \in E_x,\ v \in V_x} \psi_i(v,e) \xrightarrow{x} max.
\end{aligned}
\tag{8}
$$

On set $X$, it is necessary to select $x^0$, in which the VOF $F$ takes preferred values according to the criteria $F_i(x)$, $i = \overline{1, M + T}$. The solution of an individual multicriteria problem is the selection of a set of incomparable alternatives.

In Equations (7) and (8), the entries $\omega_i(e)$ and $\psi_i(v, e)$ denote functions of the weight and topological characteristics, respectively. In individual problems, the number of criteria is specified. In this case, $M$ sets of weights are considered to be incomparable and of a different nature. If we assume that $M$ sets are of the same type and it is possible to perform arithmetic operations on them, then sets of weights can be replaced by a single weight. For example, you can use the mean or well-known linear convolution methods and solve the problem with one set of weights.

### 2.5. Individual Formulation of the Multicriteria Problem of Placing a Multiple Center

Furthermore, an individual statement of the multicriteria problem $Z_1$ of placing a multiple center on a multi-weighted prefractal graph, an algorithm for finding solutions to particular problems are proposed, the computational complexities of the algorithms are determined, and estimates of the task criteria are calculated.

Let $x$ be a subset consisting of $p$ vertices of the set $V_L$ of the prefractal graph $G_L = (V_L, E_L)$. $d(x, v_k)$ denotes the shortest distance between vertices $x$ and $v_k \in V_L$, that is, $d(x, v_k) = \min_{v_j \in x} d(v_j, v_k)$ by a fixed set of weights from $M$. The number $s(x) = \max_{v_k \in V_L} d(x, v_k)$ is called the *separation number* of the set $x$. The set $x^*$ for which $s(x^*) = \min_{x \subseteq V_L} s(x)$ is a *multiple center* (or *p-center*) of $G_L$.

Multiple centers $\{x\}$ on $G_L$ form the set of feasible solutions $X = X(G_L) = \{x\}$.

On the SFS $x$, the VOF is determined:

$$
\begin{aligned}
F(x) &= (F_1(x), \ldots, F_M(x), \ldots, F_{2M}(x), F_{2M+1}(x), F_{2M+2}(x)), \\
F_i(x) &= s_i(x) \to \min, \ i = 1, 2, \ldots, M, \\
F_{M+i}(x) &= \textstyle\sum_{t=1}^{p} \rho_{i,t} \to \min, \ i = 1, 2, \ldots, M, \\
F_{2M+1}(x) &= h \to \min, \\
F_{2M+2}(x) &= p \to \min,
\end{aligned}
\tag{9}
$$

where $s_i(x)$ is the separation number over the $i$ set of weights; $\rho_{i,t}$ is the radius of the $t$-vertex of the $p$-center; $h$ is the number of types of centers (of different ranks); $p$ is the number of vertices of the multiple center.

All criteria in Equation (9) have a specific meaningful interpretation. The weights assigned to the edges of the prefractal graph can reflect both specific restrictions (time, distance) imposed on the system of services (emergency, fire stations, police stations, hospitals) and total costs expressed in conventional units. In practice, the separation number can mean, for example, the distance from the farthest consumer (home, organization) to the service system. The resulting value of the number $p$ of criterion $F_{2M+2}$ will be the smallest number of emergency services, and the $p$-center will be their optimal placement that meets the requirements.

Since the $M$ weights of one edge are incomparable, the SFS includes $p$-centers, not of one set of weights, but each of $M$. The criterion $F_i(x)$ minimizes the separation number, and the criterion $F_{M+i}(x)$ minimizes the sum of the radius $p$-center for each set of weights $i = 1, 2, \ldots, M$.

Figure 1 shows a prefractal graph $G_3$ with old edge adjacencies preserved. Edge weights are equal to one. The shortest path between the set $x$ and the vertex $v'$ is marked with a dotted line. The shortest distance is $d(x, v') = 4$. The separation number of $p$-center is $s(x^*) = 2$.

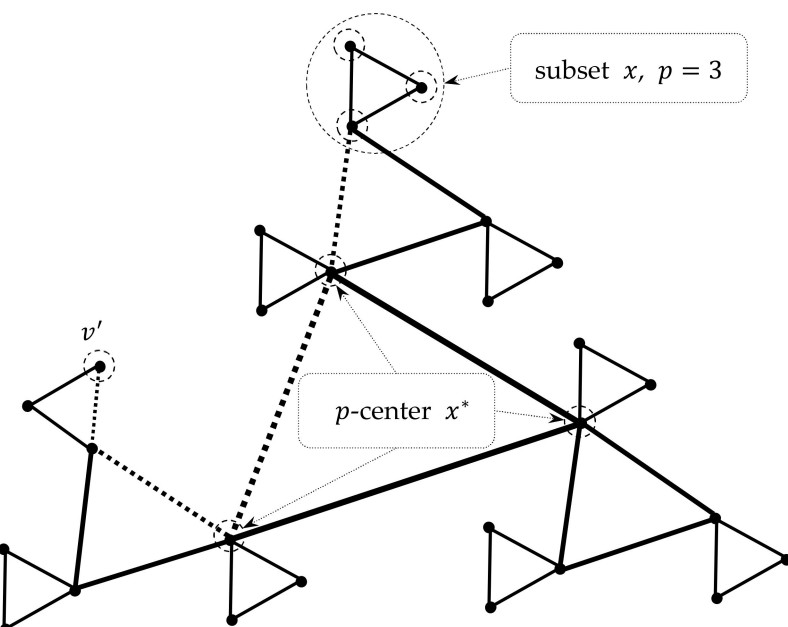

**Figure 1.** Prefractal graph $G_3$ generated by a complete 3-vertex seed, the old edges of which are adjacent.

To solve an individual problem in Equation (9), an Algorithm 1 for placing the center of a prefractal graph is proposed. Let us consider a prefractal graph $G_L$ generated by a set of seeds $H = \{H\}$ while preserving the adjacency of old edges. The edges are weighted by one set of weights, that is $M = 1$. The search for the shortest distances between vertices is carried out using Dijkstra's procedure. The shortest distance between the vertex itself is zero: $d(v, v) = 0$.

---

**Algorithm 1.** Algorithm for placing the center $(\alpha_0)$

| | |
|---|---|
| Input: | prefractal graph $G_L = (V_L, E_L)$. |
| Output: | center $x_0$ of $G_L$. |
| | for $s = 1$ to $n^{L-1}$ do: |
| 1.s. | For each common vertex $x_s^{(L)}$ of the seed $z_s^{(L)}$ find the separation number $s\left(x_s^{(L)}\right) = \max\limits_{j=1,2,\dots,n-1} d\left(x_s^{(L)}, v_j^{(L)}\right).$ |
| | for $l = L-1$ to $1$ do: |
| | for $s = 1$ to $n^{l-1}$ do: |
| $L-l+1.s.$ | For each common vertex $x_s^{(l)}$ of the seed $z_s^{(l)}$ find the separation number $s\left(x_s^{(l)}\right) = \max\limits_{j=1,2,\dots,n-1}\left(d\left(x_s^{(l)}, v_j^{(l)}\right) + s\left(x_j^{(l+1)}\right)\right).$ |
| $L+1.$ | From all vertices $x_s^{(1)}$, $s = 1, 2, \dots, n$ choose the vertex $x_0$ with the smallest separation number: $s(x_0) = \min\limits_{s=1,2,\dots,n} s\left(x_s^{(1)}\right).$ |
| **Dijkstra's procedure.** | Procedure for finding shortest distances |
| Input: | graph $G = (V, E)$. |
| Output: | shortest distances $d\left(x, v_j\right), j = 1, 2, \dots, n - 1$. |

---

As a procedure, instead of Dijkstra's algorithm, it is possible to use any other known algorithms for placing the center on a graph.

**Theorem 7.** *Algorithm $\alpha_0$ finds the center $x^*$ of the prefractal graph.*

**Proof of Theorem 7.** The algorithm $\alpha_0$ finds the center of the prefractal graph at one of the vertices of the seed $z_1^{(1)}$ of the first rank. This is facilitated by two important conditions: the first is that the adjacency of old edges is preserved, the second is that the weighting of the edges is carried out using the similarity coefficient $\theta \in (0,1)$.

Let us assume that the center of the prefractal graph is one of the vertices $v_1^{(L)}$ of the seed $z_1^{(L)}$ of rank $L$, which is incident only with new edges of rank $L$ and no old edges. The edges of $z_1^{(L)}$ are adjacent only to the edges of rank $L$. Consider the shortest path to another vertex $v_2^{(L)}$ of rank $L$, which is not in any block with vertex $v_1^{(L)}$, except for the block $B^{(L)} = G_L$. Since the adjacency of old edges is preserved, the shortest path from $v_1^{(L)}$ to $v_2^{(L)}$ will pass sequentially through the vertices of ranks $L, L-1, \ldots, 1, 2, \ldots,$ $L-1, L$. The shortest distance $d\left(v_1^{(L)}, v_2^{(L)}\right)$ will be the sum of the shortest distances between vertices of different ranks, that is: $d\left(v_1^{(L)}, v_2^{(L)}\right) = d\left(v_1^{(L)}, v^{(L-1)}\right) + d\left(v^{(L-1)}, v^{(L-2)}\right) +$ $\cdots + d\left(v^{(1)}, v^{(2)}\right) + \cdots + d\left(v^{(L-2)}, v^{(L-1)}\right) + d\left(v^{(L-1)}, v_2^{(L)}\right)$. In the case where $v_1^{(L)}$ is the center, no other vertex with a smaller separation number can be found. Since the adjacency of old edges is preserved, all shortest distances from $v_1^{(L)}$ to other vertices of $G_L$, except for the vertices of rank $L$ of $z_1^{(L)}$, pass through the vertex $v_1^{(L-1)} \in z_1^{(L)}$ of the rank $(L-1)$.

Consider the worst case where the seed is a simple cycle. Then the shortest distance from $v_1^{(L)}$ to the most distant in terms of the number of edges the vertex $v_2^{(L)}$ of the rank $L$, is equal: $d\left(v_1^{(L)}, v_2^{(L)}\right) = \theta^{L-1}a + (n-1)\theta^{L-2}a + \cdots + (n-1)\theta^1 a + (n-1)\theta^0 a +$ $(n-1)\theta^1 a + \cdots + (n-1)\theta^{L-2}a + (n-1)\theta^{L-1}a$. The shortest distance from $v_1^{(L-1)}$ to $v_2^{(L)}$ is equal: $d\left(v_1^{(L-1)}, v_2^{(L)}\right) = (n-1)\theta^{L-2}a + (n-1)\theta^{L-3}a + \cdots + (n-1)\theta^1 a + (n-1)\theta^0 a +$ $(n-1)\theta^1 a + \cdots + (n-1)\theta^{L-2}a + (n-1)\theta^{L-1}a$. We see that $d\left(v_1^{(L-1)}, v_2^{(L)}\right) < d\left(v_1^{(L)}, v_2^{(L)}\right)$ differs by exactly the weight of one edge $\theta^{L-1}a$ of the rank $L$ (see Figure 2). The shortest distance (at maximum) from $v_1^{(L-1)}$ to $v_1^{(L)}$ is equal: $d\left(v_1^{(L-1)}, v_1^{(L)}\right) = (n-1)\theta^{L-1}b$. We get $d\left(v_1^{(L-1)}, v_1^{(L)}\right) < d\left(v_1^{(L)}, v_2^{(L)}\right)$ since even $(n-1)\theta^{L-1}b < (n-1)\theta^{L-2}a$ is due to the prefractal graph weighting rule.

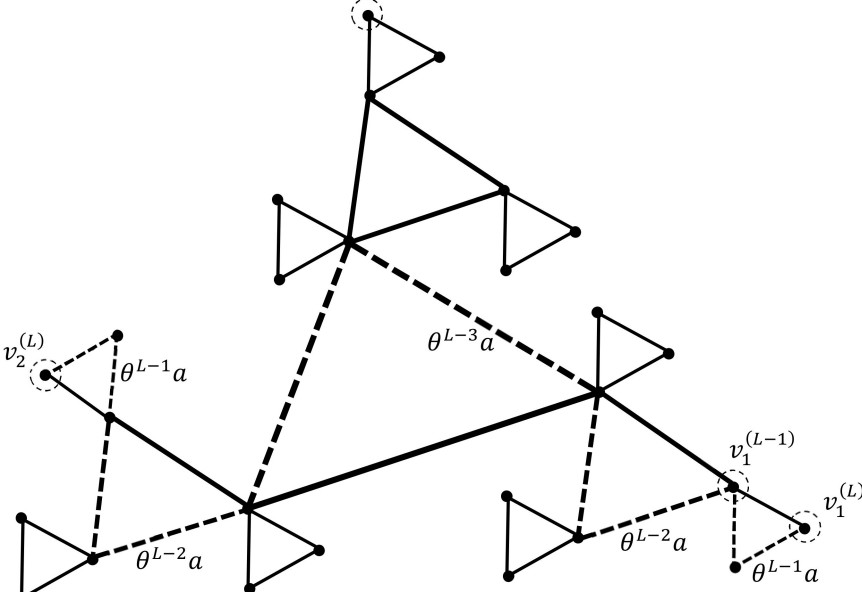

**Figure 2.** The shortest distance between two distant vertices on a prefractal graph.

Then $d\left(v_1^{(L-1)}, v_1^{(L)}\right) < d\left(v_1^{(L)}, v_2^{(L)}\right)$ and $d\left(v_1^{(L-1)}, v_2^{(L)}\right) < d\left(v_1^{(L)}, v_2^{(L)}\right)$, that is, the separation number of the vertex of $(L-1)$ rank is less than the separation number of the vertex of $L$ rank: $s\left(v^{(L-1)}\right) < s\left(v^{(L)}\right)$. Thus, no vertex of $L$ rank that is incident with new edges of $L$ rank can be the center of the prefractal graph.

In the same way, it is proved that no vertex of ranks $L-1, L-2\ldots, 2$ can be the center of a prefractal graph. Then the vertices of the first rank remain, among which the search for the center is carried out. To find the center of the prefractal graph, it is enough to find the separation numbers for each vertex of the first rank and choose the smallest among them. $\square$

**Consequence 1.** *Algorithm $\alpha_0$ takes $O(c \cdot N)$ time, where $N = n^L$ and $c = 4n^2$.*

The search for the shortest distances between all vertices in $z_s^{(L)}$ using the Dijkstra algorithm take $n^2$ operations, the choice of the maximum value take also $n^2$. The search for the separation numbers of each $z_s^{(L)}$, $s = 1, 2, \ldots, n^{L-1}$ add up to $\left(n^2 + n^2\right) \cdot n^{L-1} = 2n^{L+1}$ operations. The search for the separation numbers at all steps $l = L, L-1, \ldots, 1$ take $4n^2 \cdot N$ operations: $2n^{L+1} + 2n^L + 2n^{L-1} + \cdots + 2n^3 + n^3 + n^2 = 2\frac{n^{L+1} \cdot n - n^2}{n-1} + n^3 \leq 2n^{L+2} - n^2 + n^3 \leq 2n^{L+2} + n^3 \leq 2n^{L+2} + 2n^{L+2} = 4n^2 \cdot n^L = 4n^2 \cdot N$. At the last step are found pairwise shortest paths between all vertices $z_s^{(1)}$, $s = 1, 2, \ldots, n$ which requires $n^2 \cdot n = n^3$ operations plus time $n^2$ to find the maximum element.

**Consequence 2.** *Sequential execution of the algorithm $\alpha_0$ allows finding $x_i^*$, $i = 1, 2, \ldots, M$ center for each of $M$ set weights. Then the computational complexity of the algorithm will be $O(c \cdot N \cdot M)$.*

Determining the computational complexity of an algorithm in the worst case for intractable and complex problems is not very informative. In this case, the approach «algorithms with estimates» is used, when the quality of algorithms is evaluated using computational complexity, solution accuracy, etc.

**Theorem 8.** *Algorithm $\alpha_0$ finds the center $x^*$ on the prefractal graph, which is optimal by criteria $F_3(x^*)$ and $F_4(x^*)$, and estimated by criteria $a\frac{\theta^L - 1}{\theta - 1} \leq F_i(x^*) \leq b(n-1)\frac{\theta^L - 1}{\theta - 1} i = 1, 2$.*

**Proof of Theorem 8.** The $F_1(x)$ criterion minimizes the separation number $s_1(x)$ of the center $x^*$. In the worst case, you need to go through $(n-1)$ edges of each seed: $s_1(x^*) \leq (n-1)b + (n-1)\theta b + \cdots + (n-1)\theta^{L-1}b = b(n-1)\frac{\theta^L - 1}{\theta - 1}$. In the minimum case: $s_1(x^*) \geq a + \theta a + \cdots + \theta^{L-1}a = a\frac{\theta^L - 1}{\theta - 1}$. Then the first criterion is estimated: $a\frac{\theta^L - 1}{\theta - 1} \leq F_1(x^*) \leq b(n-1)\frac{\theta^L - 1}{\theta - 1}$. The second criterion minimizes the sum of radiuses of the multiple center. Since the multiple center consists of a single vertex, the radius is equal to the split number and is estimated: $a\frac{\theta^L - 1}{\theta - 1} \leq F_2(x^*) \leq b(n-1)\frac{\theta^L - 1}{\theta - 1}$.

Criterion $F_3(x)$ minimizes the number of types of vertices of multiple center. Since the multiple center consists of one vertex, the criterion takes its minimum possible value equal to one: $F_3(x^*) = 1$. The fourth criterion is also equal to one: $F_4(x^*) = p = 1$. $\square$

Figure 3 shows an example of finding the center of the prefractal graph $G_3$ for the case $M = 1$. The prefractal graph is weighted according to the edge weighting rule, where the initial segment is $[9; 18]$ and the weighting coefficient is $1/3$. The center search algorithm begins its work with seeds of the 3rd rank. Inside the seeds, the separation numbers of common vertices are indicated in small print.

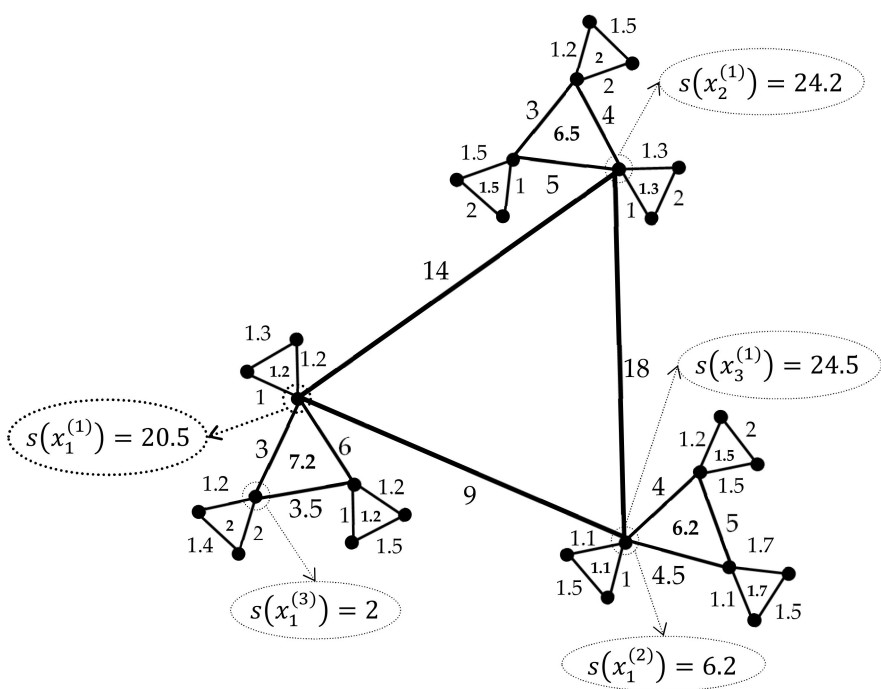

**Figure 3.** Computing the center of a weighted prefractal graph $G_3$ generated by a complete 3-vertex seed, the old edges of which are adjacent.

The center of $G_3$ is the vertex $x^* = x_1^{(1)}$, for which the separation number is minimal: $s\left(x_1^{(1)}\right) = \min(20.5; 24.2; 24.5) = 20.5$. The radius of $G_3$ is equal to the separation number of the vertex $x_1^{(1)}$: $\rho = s(x^*) = 20.5$. Criteria values $F_1(x^*) = F_2(x^*) = \rho = s(x^*) = 20.5$. The criteria estimates are as follows:

$$F_1(x^*) = F_2(x^*) \leq b(n-1) \cdot \frac{(\theta^L - 1)}{(\theta - 1)} = 18 \cdot 2 \cdot \frac{(1/3)^3 - 1}{1/3 - 1} = 52;$$
$$F_1(x^*) = F_2(x^*) \geq a(n-1) \cdot \frac{(\theta^L - 1)}{(\theta - 1)} = 9 \cdot \frac{(1/3)^3 - 1}{1/3 - 1} = 13.$$

The evaluation of the criteria is correct: $13 < F_1(x^*) = F_2(x^*) < 52$.

**Consequence 3.** *Algorithm $\alpha_0$ finds $x_i^*$, $i = 1, 2, \ldots, M$ centers of prefractal graph, which are optimal by criteria $F_{2M+1}(x_i^*)$ and $F_{2M+2}(x_i^*)$, and estimated by criteria $a_i \frac{\theta^L - 1}{\theta - 1} \leq F_i(x_i^*) \leq b_i(n-1) \frac{\theta^L - 1}{\theta - 1}$, $a_i \frac{\theta^L - 1}{\theta - 1} \leq F_{M+i}(x_i^*) \leq b_i(n-1) \frac{\theta^L - 1}{\theta - 1}$, $i = 1, 2, \ldots, M$.*

In this case, it is assumed that the prefractal graph is weighted by $M$ weights, where $M > 1$. Each center $x_i^*$ is found by a set of weights $i = 1, 2, \ldots, M$. The criterion $F_i(x_i^*)$ gives an estimate for the separation number $s_i(x_i^*)$ of a particular center $x_i^*$. For the criterion $F_{M+i}(x_i^*)$ an estimate of the radius of each center $x_i^*$, $i = 1, 2, \ldots, M$ are presented.

All presented solutions $x_i^*$ of the individual problem of placing a multiple center belong to the set of feasible solutions $x_i^* \in X$. The solutions $x_i^*$ are Pareto-optimal solutions and are included in the Pareto set $x_i^* \in \widetilde{X}$. Moreover, among $x_i^*$ there are identical ones. The number of different solutions cannot be more than $n$—the number of seed vertices.

## 3. Results and Discussion

Setting an optimization problem on a graph implies using the graph as a tool for modeling some object or process (the structure of a social network [40–42], transport and logistics systems [43], processes in cryptocurrency networks [44,45], DNA structure, etc.). The solution of such a problem is a subgraph of a given graph with optimal values of parameters or characteristics. For example, selecting a subgraph of maximum or minimum

weight, finding the shortest paths, selecting the most stable subgraph, etc. The choice of the parameter to be optimized is determined by the criterion, and the solution lies in the area of feasible solutions—all possible subgraphs corresponding to the given definition of the solution.

In this paper, we consider the class of prefractal graphs on which multicriteria optimization problems are studied. Brief definitions and notation of a prefractal graph are considered, as well as some important characteristics used in theorems and proofs. Multicriteria settings differ from single-criteria ones by their laboriousness and complexity. Nevertheless, for modeling and solving multiparameter problems in the economy and other areas of human life, it is advisable to use precise multicriteria statements. First of all, the definitions of the area of feasible solutions and their representative samples are given— the set of alternatives (Pareto, complete, lexicographic). Common individual problems are divided into separate classes ($Z_1$–$Z_{11}$), which describe solutions and sets of feasible solutions. A special type of classification by structural affiliation is also proposed, which is represented by two main classes of problems: $Z_{l_1}^{l_2}$ and $B_{l_1}^{l_2}$.

Complete problems on graphs are considered separately. A problem in which the set of feasible solutions $X$ coincides with the Pareto set $\widetilde{X}$ and the complete set of alternatives $X^0$: $X = \widetilde{X} = X^0$ is a complete problem. That is, the general solution of such a problem is to select the entire set of alternatives $X$.

The paper presents well-known Lemmas 1–3 and Theorems 1–6 that are also true for the class of prefractal graphs. Lemma 1 defines the relationship between the cardinality of the complete set $X^0$ and the cardinality of the set of values of the objective function $F\left(\widetilde{X}\right)$ of the Pareto set: $\left|X^0\right| = \left|F\left(\widetilde{X}\right)\right|$. Lemma 2 says that increasing the criteria of the vector-objective function does not reduce the Pareto set and the complete set of alternatives. For a fixed value of the index $t$, the family of individual problems $Z_t$ can contain both complete and incomplete problems (Lemma 3). For example, problem $Z_1$ of identifying a multiple center will be complete under some criteria, but not under others. Theorems 4-7 indicate the completeness conditions for individual problems $Z_3$–$Z_5$, $Z_7$—the vector-objective function contains at least two weight criteria. That is, the inclusion of many topological criteria does not mean the completeness of the problem, but the presence of at least two weight criteria guarantees completeness. Theorem 5 says that for multicriteria problems with two or more weight criteria, linear convolution methods are not applicable, both in the sense of convolution of weights and the criteria themselves. In some cases, linear convolution algorithms are applicable to search for a lexicographic set of alternatives of a multicriteria integer problem (Theorem 6).

The main result to which attention should be paid is the general formulation of a multicriteria problem on a (multi-weighted) prefractal graph. The vector-objective function is represented by two types of criteria, weight and topological. Standard forms are presented for each type of criteria. As an example, an individual multicriteria problem of identifying a multiple center of an $M$-weighted prefractal graph with two types of weight criteria and two types of topological criteria is formulated. Algorithm $\alpha_0$ is proposed for solving an individual problem. Theorem 7 on the validity of the algorithm is formulated and proved. The execution time of the algorithm $\alpha_0$ depends linearly on the dimension of the problem $N$, as well as on the parameter $c$, which depends on the number of seed vertices. The $c$ parameter is a constant since it is determined before the start of the algorithm and does not change during its execution. The proposed algorithm $\alpha_0$ is an «algorithm with estimates». Algorithm $\alpha_0$ finds one possible center $x^*$ and offers the following criteria estimates. Two criteria $F_3$ and $F_4$ is optimal (with minimum values): $F_3(x^*) = F_4(x^*) = 1$, and for two weight criteria $F_1$, $F_2$ estimates are calculated: $a\frac{\theta^L - 1}{\theta - 1} \leq F_i(x^*) \leq b(n-1)\frac{\theta^L - 1}{\theta - 1}$, $i = 1, 2$. As an example, the algorithm is applied on a prefractal graph $G_3$ to find the center. Exact values were calculated for all criteria and compared with estimates.

Consequence 3 should be noted separately. The execution of the algorithm $\alpha_0$ for each weight $i = 1, 2, \ldots, M$ allows us to find the centers of $x_i^*$. The centers $x_i^*$ are Pareto optimal solutions: $x_i^* \in \widetilde{X}$.

A single-criteria task differs from a multicriteria one by the presence of many alternatives. Finding the optimal solution to a multicriteria problem is often a laborious process. Moreover, in some problems it is difficult to find at least some solution or describe the set of alternatives.

In this paper, we give a definition of a complete problem for which the set of alternatives coincides with the Pareto set. That is, any solution found from the set of feasible solutions is Pareto optimal. The proposed classification of individual problems $Z_1$–$Z_{11}$ is necessary for further study of their sets of feasible solutions, in particular under what conditions these problems become complete. Theorems 1–4 suggest such completeness conditions for problems $Z_3$–$Z_5$, $Z_7$. Then any solution found will be optimal. For other tasks, it is necessary to formulate proofs and highlight such conditions, and it is possible that they will be repeated.

Since we are talking about several weight criteria that cannot be collapsed into a single one by a linear convolution method, the paper proposes a formulation of a general multicriteria problem with two types of criteria—weight and topological. In graph theory, each weight criterion is usually given its own set of weights, that is, we are talking about a multi-weighted graph. For this purpose, in Section 2.4, a rule for weighing a prefractal graph by many real numbers is proposed, and standard forms of criteria are also presented.

For the class $Z_1$, an individual formulation of the problem is proposed. This paper does not provide a theorem and a proof of the completeness of this problem, therefore, up to this point, for the proposed Algorithm 1, it cannot be said that it finds a Pareto optimal solution. On the other hand, Algorithm 1 already finds the lexicographic optimum (according to one criteria) and estimates are calculated for all criteria (within what limits the values of these criteria lie).

Algorithm 1, denoted as $\alpha_0$, is a reference algorithm, on the basis of which it is possible to construct algorithms for any problem from the class $Z_1$. It should also be noted once again that Dijkstra's algorithm is used as a procedure which can be changed to any modern algorithm for finding the shortest chains with better computational characteristics. This will reduce the parameter c of the $O(c \cdot N)$ estimate of the execution time of the algorithm.

The result of future research should be the selection of completeness conditions for all problems $Z_1$–$Z_{11}$ and the development of families of algorithms for covering each class $Z_1$–$Z_{11}$. Separately, the possibility of enumerating or at least limiting the sets of alternatives of these tasks should be studied. Also, additional studies require the multiweightedness of prefractal graphs, including fuzzy weights.

This work is theoretical in nature. In the future I plan to present the results of using the apparatus of prefractal graphs for modeling large-scale objects and processes, in particular of social networks and large graphs.

## 4. Conclusions

In this paper, we propose an introduction to multicriteria discrete optimization for a special class of prefractal graphs. An individual multicriteria problem is determined by the set of its feasible solutions. Therefore, for well-known graph problems, the corresponding classes of multicriteria problems on prefractal graphs ($Z_1$–$Z_{11}$) are proposed. Further research is expected to expand this list of tasks. For some problems, the conditions for their completeness ($Z_3$–$Z_5$, $Z_7$) are proposed, but it is necessary to consider other problems, including those with many weight and topological criteria. It is also possible to expand the types of weight and topological criteria.

In modern problems, in addition to the multiplicity of sets of weights, non-deterministic weights are considered. These include, for example, interval and fuzzy numbers, time series, and other types of uncertainty. Therefore, it is necessary to consider sets of weights not only with real numbers but also non-deterministic ones.

The prefractal graph also belongs to the class of dynamic graphs and is represented by its trajectory $G_1, G_2, \ldots, G_L$ [46–48]. The proposed formulations of multicriteria problems refer to the graph $G_L$ from the trajectory. It is necessary to consider the applicability of the statements to the sequence $G_1, G_2, \ldots, G_L$ and propose a through operation of the algorithms and link the sequence of solutions.

It should be noted that prefractal graphs are used to model large graphs and networks. For a number of artificial networks, it is difficult to talk about the exact construction of a prefractal graph as a model. In this case, it is necessary to apply a measure of similarity and then use it in all calculations as an indicator of the error in solving the problem.

To speed up algorithms, one should also pay attention to the possibility of parallel implementation of sequential algorithms. The structure of prefractal graphs makes it possible to parallelize many well-known sequential algorithms [49].

Special attention should be paid to the stability of solutions, both in the case of structural changes and changes in weight values. This is especially true for the dynamic properties of prefractal graphs.

**Funding:** This research was funded by Financial University under the Government of the Russian Federation.

**Institutional Review Board Statement:** Not applicable.

**Informed Consent Statement:** Not applicable.

**Data Availability Statement:** Not applicable.

**Conflicts of Interest:** The author declares that they have no conflict of interest.

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
