# Peer review of "Multicriteria Optimization Problem on Prefractal Graph"

_mathematics, doi:10.3390/math10060930_

Round 1
Reviewer 1 Report
First of all, this is a well–designed and written study, however, it is similar to or some kind of extension of the author’s previous study, i.e. " Research of NP-Complete Problems in the Class of Prefractal Graphs" in Oct.2021.
Here, the author declares that several formulations of a multicriteria problem that are stated as weight, degree, and diameter, the «capacity» of edges on lines 187-190 are discussed. These multicriteria problems are tried to be optimized or solved on multi-weighted (with real numbers) prefractal Z0-Z11 graphs. Anyway, it is believed to be hard to implement for real-life problems. The main proposal is a multicriteria problem on multi-weighted prefractal graphs. The problems are solved in Consequences 1-3 and based on Аlgorithm α0 in which those proposed criteria are used. Dijkstra's approach can be used for fundamental cases such as prefractal graphs and is suitable for teaching the students shortest path problems, but this approach may not be suitable for real-life problems. So, I could not notice a big contribution to the field.
Author Response
Dear reviewer, thank you for your valuable comments and the opportunity to further improve the scientific work.
Response to Reviewer 1 Comments
Point 1: First of all, this is a well–designed and written study, however, it is similar to or some kind of extension of the author’s previous study, i.e. " Research of NP-Complete Problems in the Class of Prefractal Graphs" in Oct.2021.
Response 1: Indeed, this work is a continuation of a series of articles about prefractal graphs. But prefractal graphs are more of a tool for solving certain problems. In the first article, this is the allocation of conditions for the solvability of some NP-complete problems due to the special properties of prefractal graphs. In the current article, this is the formulation of multicriteria problems on the class of prefractal graphs and the possibilities for developing polynomial algorithms for finding Pareto or lexicographic solutions.
Point 2: Here, the author declares that several formulations of a multicriteria problem that are stated as weight, degree, and diameter, the «capacity» of edges on lines 187-190 are discussed. These multicriteria problems are tried to be optimized or solved on multi-weighted (with real numbers) prefractal Z0-Z11 graphs. Anyway, it is believed to be hard to implement for real-life problems..
Response 2: An example of selecting a spanning tree according to different criteria was given in order to clarify that multicriteria problems are classified according to the area of feasible solutions. That is, for all multicriteria problems of selecting spanning trees, the set of feasible solutions is the same, but a specific optimal solution is found from this set according to a given criterion.
The following is a list of all the main problems that can be combined over sets of feasible solutions. Each of the tasks Z1-Z11 is in fact a class of individual tasks.
Prefractal graphs are a tool for modeling real world networks, such as social networks, transport and logistics networks, cryptocurrency networks, a network of large-scale clustering of matter in the universe, and many others. Any tree graph is recognized as a prefractal graph (another author's theorem). In the modeling of real networks, a lot of weights (of one edge) are always involved, and a multicriteria problem immediately arises (with many weight criteria). For example, the problem of optimal placement of towers of a dynamic wireless communication network.
Point 3: The main proposal is a multicriteria problem on multi-weighted prefractal graphs. The problems are solved in Consequences 1-3 and based on Аlgorithm α0 in which those proposed criteria are used. Dijkstra's approach can be used for fundamental cases such as prefractal graphs and is suitable for teaching the students shortest path problems, but this approach may not be suitable for real-life problems. So, I could not notice a big contribution to the field.
Response 1: The article proposes an apparatus for multiobjective optimization on prefractal graphs. And prefractal graphs are a tool for modeling real networks of large dimensions.
As an example, the multiobjective problem of selecting a multiple center and the algorithm α0 are given. Dijkstra's algorithm is used as a procedure, but as stated in the article, the procedure can be replaced by other modern shortest path algorithms. Algorithm α0 is a prefabricated algorithm that can use other well-known algorithms in the form of procedures. Also an important result is the evaluation of the task criteria for the algorithm. After the completion of the algorithm, the values of the criteria will be guaranteed to be located on the specified segments (or take exact values). The article proposes an approach for developing algorithms with guaranteed estimates for all classes of proposed problems Z1-Z11.

Reviewer 2 Report
This paper intends to explore multicriteria optimization problems on prefractal graph, and consider individual problem of placing a multiple center with two types of weight criteria and two types of topological ones. In general, the main idea is interesting and sound, however several issues should be further addressed:
- Some concise background of multicriteria optimization problems as well as study gaps should be added in abstract.
- Literature review in the current version is not systematic, a separate section should be added titled related works.
- Main study motivations and innovation points should be listed in Section 1.
- Some practical implications in terms of the presented theorems should be discussed.
- What is the study limitations, they need to be included in the last section.
- Language quality need to be better refined to meet the journal standard.
Author Response
Point 1: Some concise background of multicriteria optimization problems as well as study gaps should be added in abstract.
Response 1: Added text to the annotation:
Currently, priority is given to the study of applied multicriteria problems with specific criteria; there is no classification of criteria according to their type and content. There are few studies with fuzzy criteria, both weight and topological. Little attention is paid to the stability of solutions, and this is necessary when modeling real processes due to their dynamism. It is also necessary to study the behavior of solution sets for various general and individual problems. The theory of multicriteria optimization is a rather young branch of science and requires the development of not only particular methods, but also the construction of a methodological basis. This is also true in terms of discrete graph-theoretic optimization..
Point 2: Literature review in the current version is not systematic, a separate section should be added titled related works..
Response 2: The introduction is divided into three subsections, including the subsection related works, added several additional publications in this subsection:
- Statnikov, R.; Bordetsky, A.; Matusov, J.; Sobol’, I.; Statnikov, A. Definition of the feasible solution set in multicriteria opti-mization problems with continuous, discrete, and mixed design variables. Nonlinear Analysis: Theory, Methods & Applications 2009, 71 (12), e109-e117. https://doi.org/10.1016/j.na.2008.10.050
- Puerto, J.; Rodríguez-Chía, A.M. Quasiconvex constrained multicriteria continuous location problems: Structure of non-dominated solution sets. Computers & Operations Research 2008, 35 (3), 750-765. https://doi.org/10.1016/j.cor.2006.05.002.
Point 3: Main study motivations and innovation points should be listed in Section 1.
Response 3: Added text to section 1.
Thus, in this paper, we propose a toolkit of prefractal graphs for formulating mul-ticriteria problems. A rule for weighing a prefractal graph by many real numbers is pro-posed, and classes of individual problems with common sets of feasible solutions are presented as an example. Two types of criteria are proposed - topological and weight.
Point 4: Some practical implications in terms of the presented theorems should be discussed.
Response 4: Added short conclusions on theorems.
The completeness of the problem means that any found solution from the set of feasi-ble solutions will be Paretо and at the same time be included in the complete set of alter-natives. Moreover, singling out the entire set of alternatives means singling out the entire Pareto set. Theorems 4-7 give completeness conditions for problems under which any so-lution from the set of alternatives can be found and it will be Pareto.
Theorem 8 postulates the fact that in the case of 2 or more weights, the problem is unsolvable using linear convolution algorithms. In this sense, it is necessary to consider weights of different nature, incomparable among themselves.
Point 5: What is the study limitations, they need to be included in the last section.
Response 5: Added a short text in the last section.
It should be noted that prefractal graphs are used to model large graphs and networks. For a number of artificial networks, it is difficult to talk about the exact construction of a prefractal graph as a model. In this case, it is necessary to apply a measure of similarity and then use it in all calculations as an indicator of the error in solving the problem.
Point 6: Language quality need to be better refined to meet the journal standard.
Response 6: I reviewed the text again for grammatical errors and typos, corrected as much as possible.

Reviewer 3 Report
The author of the paper entitled “ Multicriteria Optimization Problem on Prefractal Graph ” presented general formulation of a multicriteria problem on a multi-weighted prefractal graph; defines three sets of alternatives Pareto, complete and lexicographic; and proposes a classification of individual problems according to the set of feasible solutions. As an example, the author considered an individual problem of placing a multiple center with two types of weight criteria and two types of topological ones. An algorithm with estimates of all criteria of the problem is proposed.
The paper idea is good, it presents an acceptable contribution, and it can be further processing after major and essential revisions, as follows:
1- first, the abstract is somewhat messy. I think you should elaborate your method in the abstract supported by chivied results (numerical results). Avoid long background and unnecessary description in the abstract.
2- The first time you use an acronym in the text, please write the full length name and the acronym in parenthesis. Do not use acronyms in the headings and highlights.
3- Carefully re-check Algorithm ??. Also, please care about the numbering of the algorithms and follow journal format.
4-Analysis of the results is missing in the paper. There is a big gap between the results and conclusion. You have to analyze the results and relate them to the structure of all theorems and proofs. So, adding more detailed description is required.
5- The results should be further elaborated to show how they could be used for the real applications.
Author Response
Response to Reviewer 3 Comments
Point 1: 1- first, the abstract is somewhat messy. I think you should elaborate your method in the abstract supported by chivied results (numerical results). Avoid long background and unnecessary description in the abstract.
Response 1: Some edits have been made to the abstract. Other reviewers, on the contrary, asked to expand the abstract and add details..
Point 2: 2- The first time you use an acronym in the text, please write the full length name and the acronym in parenthesis. Do not use acronyms in the headings and highlights.
Response 2: Checked all headings and removed abbreviations. In some cases, full length name is provided to reinforce the text.
Point 3: 3- Carefully re-check Algorithm ??. Also, please care about the numbering of the algorithms and follow journal format.
Response 3: I rechecked the algorithm, added the numbering "1". The algorithm is denoted as ?0, since further it is necessary to talk about the family of algorithms ?={?0, ?1,…, ?L-1} for solving the individual problems Z1.
Point 4: 4-Analysis of the results is missing in the paper. There is a big gap between the results and conclusion. You have to analyze the results and relate them to the structure of all theorems and proofs. So, adding more detailed description is required.
Response 4: Added more detailed description.
A single-criteria task differs from a multicriteria one by the presence of many alternatives. Finding the optimal solution to a multicriteria problem is often a laborious process. Moreover, in some problems it is difficult to find at least some solution or describe the set of alternatives.
In this paper, we give a definition of a complete problem for which the set of alternatives coincides with the Pareto set. That is, any solution found from the set of feasible solutions is Pareto optimal. The proposed classification of individual problems - is necessary for further study of their sets of feasible solutions, in particular, under what conditions these problems become complete. Theorems 4-7 suggest such completeness conditions for problems - , . Then any solution found will be optimal. For other tasks, it is necessary to formulate proofs and highlight such conditions, it is possible that they will be repeated.
Since we are talking about several weight criteria that cannot be collapsed into a single one by linear convolution method, the paper proposes a formulation of a general multicriteria problem with two types of criteria - weight and topological. In graph theory, each weight criterion is usually given its own set of weights, that is, we are talking about a multi-weighted graph. For this purpose, in subsection 2.4, a rule for weighing a prefractal graph by many real numbers is proposed, and standard forms of criteria are also presented.
For the class , an individual formulation of the problem is proposed. This paper does not provide a theorem and a proof of the completeness of this problem, therefore, up to this point, for the proposed Algorithm 1, it cannot be said that it finds a Pareto optimal solution. On the other hand, Algorithm 1 already finds the lexicographic optimum (according to one criteria) and estimates are calculated for all criteria (within what limits the values of these criteria lie).
Algorithm 1, denoted as , is a reference algorithm, on the basis of which it is possible to construct algorithms for any problem from the class . It should also be noted once again that Dijkstra's algorithm is used as a procedure, which can be changed to any modern algorithm for finding the shortest chains with better computational characteristics. This will reduce the parameter of the estimate of the execution time of the algorithm.
The result of future research should be the selection of completeness conditions for all problems - , the development of families of algorithms for covering each class - . Separately, the possibility of enumerating or at least limiting the sets of alternatives of these tasks should be studied. Also, additional studies require the multiweightedness of prefractal graphs, including fuzzy weights.
Point 5: 5- The results should be further elaborated to show how they could be used for the real applications.
Response 5: This work is theoretical in nature, in the future it is planned to present the results of using the apparatus of prefractal graphs for modeling large-scale objects and processes, in particular social networks and large graphs.
Research is already underway and there are first applied results. A new article is being prepared.
Also, additional materials on the comments of other reviewers have been added to the text of the article, highlighted in color.

Reviewer 4 Report
In this paper, the classes of multicriteria problems on prefractal graphs (?1-?11) are proposed. For some problems, the conditions for their completeness (?3-?5, ?7) are proposed. A general formulation of a multicriteria problem on a multi-weighted prefractal graph is proposed, three sets of alternatives - Pareto, complete and lexicographic are defined, and a classification of individual problems according to the set of feasible solutions is proposed. An algorithm with estimates of all criteria of the problem is proposed.
I find the paper interesting, well-written, easy-to-follow, and English is good. I believe that this paper can be accepted in the current form.
Author Response
Thank you for your positive feedback. Also, some materials were added to the text of the article on the recommendation of other reviewers. Highlighted in color.
Round 2
Reviewer 2 Report
Authors have revised the manuscript as per the comments of the reviewers.Reviewer 3 Report
The author addressed all of my comments raised in the previous round.
I think the paper is ready for publication.